# Chromatographic Method for Monitoring of Pesticide Residues and Risk Assessment for Herbal Decoctions Used in Traditional Korean Medicine Clinics

**DOI:** 10.3390/molecules28083343

**Published:** 2023-04-10

**Authors:** Se-Mi Kang, Jae-Hee Won, Ji-Eun Han, Jong-Hyun Kim, Kyeong-Han Kim, Hye-In Jeong, Soo-Hyun Sung

**Affiliations:** 1Department of Public Infrastructure Operation, National Institute of Korean Medicine Development, Seoul 04554, Republic of Korea; 2Department of Policy Development, National Institute of Korean Medicine Development, Seoul 04554, Republic of Korea; 3Department of Medical Classics and History, College of Korean Medicine, Gachon University, Seongnam-si 13120, Republic of Korea; 4Department of Preventive Medicine, College of Korean Medicine, Woosuk University, Jeonju-si 54986, Republic of Korea; 5Department of Preventive Medicine, College of Korean Medicine, Kyung Hee University, Seoul 02447, Republic of Korea

**Keywords:** herbal decoction, traditional Korean medicine, pesticide residues, risk assessment

## Abstract

The presence of pesticide residues in herbs and the herbal products derived from them raises serious health concerns. This study was conducted to investigate the residual pesticide concentrations and assess potential human health risks from herbal medicines used in traditional Korean medicine clinics. A total of 40 samples of herbal decoctions were collected from 10 external herbal dispensaries. The pesticide residues were analyzed by the multiresidue method for 320 different pesticides using liquid chromatography tandem mass spectrometry (LC-MS/MS) and gas chromatography tandem mass spectrometry (GC-MS/MS). As a result of the monitoring, carbendazim was detected at 0.01 and 0.03 μg/g in eight samples and no pesticide was detected in the other herbal decoctions. Carbendazim was set for each individual item as less than 0.05 μg/g in Paeoniae radix, less than 0.05 μg/g in Cassiae semen, less than 2.0 μg/g in Lycii fructus, and less than 10 μg/g in Schisandrae fructus (dried). Therefore, the results of this study suggested that the detected pesticide residues in herbal decoctions could not be considered as posing a serious health risk.

## 1. Introduction

In a rapidly growing population, various pesticides are used to increase overall agricultural production [1]. Because pesticides tend to remain in harvested crops and these residues can harm humans, many countries use a variety of methods to detect pesticide residues [2]. Lists of pesticides that require testing in accordance with the pharmacopoeia of each country and the national food safety standards have been established [3,4]. Moreover, there has been a growing interest in analyzing more pesticides and developing new pretreatment methods [5,6].

Herbal medicines are grown as agricultural products before being washed, cut, dried, and packaged in facilities with good manufacturing practices; therefore, they are exposed to various pesticides [7]. In Korea, Monograph Part 2 of the Korean Pharmacopoeia presents individual standard specifications for herbal medicines and preparations [8]. Standards for pesticide residues, including α-BHC, β-BHC, δ-BHC, γ-BHC, aldrin, dieldrin, endrin, P.P′-DDD, P.P′-DDE, O.P′-DDT, and P.P′-DDT, ranging from 11 to 31, are set and presented depending on the product. The test method involves using gas chromatography equipped with an electron capture detector, nitrogen-phosphorus detector, etc., or high-performance liquid chromatography (HPLC) after pretreatment.

However, the number of pesticides registered every year for the cultivation of agricultural products is increasing [9] because herbal medicines are imported from various countries, and it is becoming difficult to secure the safety of herbal medicines using only the herbal medicine test method No. 30 and the standards in the Korean Pharmacopoeia. In particular, in the case of herbal decoctions, since patients take them within approximately 24 h after dispensing, test results should be obtained as quickly as possible [10]. However, various types of herbal medicines are used according to the prescriptions, and the procedure to analyze pesticide residues in the Korean Pharmacopoeia is so complicated that it takes approximately 2–7 days to determine the results [11]. Therefore, developing an effective analysis method that can shorten the test time and increase the accuracy is necessary.

In Korea, the QuEChERS sampling method has been applied to the preprocessing stage in the food sector for a long time, and many pesticide multicomponent analysis methods have been developed using HPLC, gas chromatography (GC), gas chromatography tandem mass spectrometry (GC-MS/MS), and liquid chromatography tandem mass spectrometry (LC-MS/MS) [12]. Representatively, there are multiclass pesticide multiresidue methods [13] in the Korea Food Code and methods of analyzing harmful substances, such as the agricultural products of the Agricultural Products Quality Management Service [14]. In this study, the pretreatment process was supplemented with a single test method to overcome the limitations of human resources and time, in accordance with the limitations of the analysis of pesticide components in the existing analysis methods (multiclass pesticide multiresidue methods), the limitations of the diversity of pretreatment methods, and instrumental analysis. The specificity, linearity, accuracy, detection limit, and quantification limit of the pesticide analysis methods LC-MS/MS and GC-MS/MS were verified, and the pesticide detection results of 40 prepared decoctions using the test methods are presented.

## 2. Results

### 2.1. Test Analysis Method Verification Result

#### 2.1.1. Specificity

GC-MS/MS Target Pesticide

The specificity was confirmed under multiple reaction monitoring (MRM) conditions for 113 standard products for GC-MS/MS analysis to confirm the selectivity of the measured analytes without interference from other components (Figure 1).

LC-MS/MS Target Pesticide

To confirm the selectivity to measure analytes without interference from other components, specificity was confirmed under MRM conditions for 207 standard products for LC-MS/MS analysis (Figure 2).

#### 2.1.2. Linearity

The linearity between instrument signals according to pesticide concentration was evaluated in the dilution range of 1−200 μg/kg of the calibration curve standard solution. As a result of examining the linearity of 113 diluted pesticide mixture standard solutions for GC-MS/MS analysis in 5 stages at concentrations of 5, 10, 20, 100, and 200 μg/kg, most showed good linearity with R^2^ ≥ 0.99. As a result of examining the linearity of the 207 diluted standard pesticide mixtures for LC-MS/MS analysis at concentrations of 1, 5, 10, 100, and 200 μg/kg in 5 stages, all showed good linearity with R^2^ ≥ 0.99 (Appendix A).

#### 2.1.3. Accuracy

Accuracy, the degree of agreement between the measurement result and the standard value, was measured by recovery. The accuracy of the simultaneous analysis of pesticide residues in the decoction was confirmed by adding a standard solution to Galgeun-tang. For the recovery rate test, standard solutions were added to Galgeun-tang at concentrations of 10, 50, and 100 μg/kg, extracted using the above pretreatment method, and the test solution was then analyzed using LC-MS/MS and GC-MS/MS. The recovery rate results are shown in Appendix A (See the Appendix A). Galgeun-tang is the herbal decoction most commonly prescribed to patients in traditional Korean medicine (TKM) and was selected after expert consultation to measure accuracy [15]. Herbal medicines (*Pueraria lobata* Ohwi, *Cinnamomum cassia* Presl, *Ephedra sinica* Stapf, *Paeonia lactiflora* Pallas, *Glycyrrhiza uralensis* Fischer, *Zingiber officinale* Roscoe, and *Zizyphus jujuba* Miller) that make up Galgeun-tang are all manufactured in an hGMP manufacturing facility licensed by the Korea Ministry of Food and Drug Safety [16,17].

The recovery rates of the 207 pesticide components subjected to LC-MS/MS analysis were 65−161% in the case of low concentrations; 201 pesticides with standard deviations within 15% consisted of 201 species; and 201 components were qualitative. Six types of pesticides, cyazofamid, cyflufenamid, fenoxaprop-ethyl, gibberellic acid, propaquizafop, and pyroquilon, were outside the recovery rate of 70−125%. The recovery rates at high concentrations were 79−119%, and the standard deviation was within 15% for the 207 pesticides, all of which could be analyzed at high concentrations.

The recovery rates of the 113 pesticide components analyzed by GC-MS/MS were 90−1349% at low concentrations. There were 111 pesticides whose recovery rates were 70−125%, with standard deviations within 15%, and there were 111 types of ingredients that could be quantified. The recovery rate was higher than 125% for two types of pesticides (prochloraz and indanofan).

The recovery rates at high concentrations were 97−189%. The recovery rates of 112 pesticides were 70−125%; the standard deviation was within 15%, and 112 components were available for qualitative treatment. Indanofan had a recovery rate of 189%, and the standard deviation was 20%, making it qualitatively difficult.

#### 2.1.4. Limit of Detection and Quantification

The detection and quantification limits were calculated using the Mass Hunter program (version 11.2; Agilent Technologies, Santa Clara, CA, USA) after repeated measurements of the lowest concentration seven times based on the calibration curve prepared to confirm linearity. It was calculated using the slope of the calibration curve and the deviation of repeated measurements. The limit of detection (LOD) of each pesticide was 0.05 to 5.0 μg/kg in the LC-MS/MS analysis and 0.2 to 7.0 μg/kg in the GC-MS/MS analysis. The limit of quantification was in the range of 0.15 to 15 μg/kg in the LC-MS/MS analysis and 0.6 to 20 μg/kg in the GC-MS/MS, and trace amounts of pesticide components could be detected at the level of 10 μg/kg contained in the sample. 

### 2.2. Pesticide Residues in Analyzed Samples

As a result of the analysis of 40 herbal decoctions prepared in the outpatient bathroom, no pesticides other than carbendazim were detected. Carbendazim was detected in the range of 0.01 to 0.03 μg/g in 8 samples of 40 herbal decoctions.

## 3. Discussion

The existing method (QuEChERS sample pretreatment method) is a test method for solids such as food and agricultural products, and there is a difference in the detection concentration depending on the volume of solids and the type of sample. Therefore, most tests using the existing method are used only as monitoring test methods (detection checks), and the quantitative results at the time of detection are obtained by conducting individual experiments for each pesticide component. However, it was confirmed that this test method is a stable method with accuracy and precision that can be widely applied without large deviations for liquid samples with certain properties, such as the decoction obtained by first hot water extraction of the sample. This study presented an improved test method that can be applied to liquid types with similar properties to decoction samples and that is valuable in that accuracy and precision were verified through validation.

The possibility of simultaneous multicomponent analysis of 320 pesticides using LC-MS/MS and GC-MS/MS was investigated to analyze the pesticide residues in herbal decoctions. The validity of the test method was verified by applying a preprocessing method modified from the existing QuEChERS method.

In AOAC and Codex, the suitability of the analysis method used in a study is judged by a 70−125% recovery rate and a 15% relative standard deviation. The recovery rate of the analytical method using LC-MS/MS and GC-MS/MS was at least 70%, and the relative standard deviation was less than 15%, meeting international standards, while eight pesticides at 10 μg/kg concentration, four pesticides at 50 μg/kg concentration, and one pesticide at 100 μg/kg concentration were excluded.

Therefore, the analysis method applied in this study is considered applicable to the analysis of multicomponent pesticides remaining in herbal decoctions.

Among the 320 species to be analyzed, it is considered necessary to apply and develop additional test methods to improve the preprocessing methods and increase the efficiency of the analysis of indanofan components that cannot be quantified beyond the range of recovery rates [18].

The multicomponent analysis method for various pesticides using LC-MS/MS and GC-MS/MS reviewed in this study is expected to be applicable for monitoring herbal decoctions.

Currently, no distinct regulations exist for managing harmful substances, including residual pesticides, in the herbal decoctions utilized by TKM clinics. This is due to the fact that such decoctions are not classified as products approved by the MFDS but rather as traditional medicines prepared through herb decoction at TKM clinics. Nonetheless, the Herbal Medicine Test Method of the Korean Pharmacopoeia applies specific residue limits for various pesticides in herbal medicines and extracts, including total DDT (sum of p,p′-DDD, p,p′-DDE, o,p′-DDT, and p,p′-DDT) at 0.1 ppm or less, dieldrin at 0.01 ppm or less, total BHC (the sum of α, β, γ, and δ-BHC) at 0.2 ppm or less, aldrin at 0.01 ppm or less, and endrin at 0.01 ppm or less [8]. Furthermore, the Korean Pharmacopoeia applies an interim standard of 0.01 ppm or less for pesticides that lack established standards in the herbal medicine test method in processed foods [8]. Although this study’s validation of the test method demonstrated that all LODs were below 0.01 ppm, the LOQs for five pesticides, including bifenox, exceeded 0.01 ppm, which is higher than the residual pesticide limits set by the Korea MFDS for herbal extracts and processed foods.

Carbendazim is commonly used to control fungal diseases in vegetables and fruits [19,20]. It was used worldwide before toxicological evidence was detected, and the U.S. Environmental Protection Agency canceled the registration. Although it has carcinogenicity and reproductive toxicity and can damage organs, such as the liver, it is still used in agriculture in some countries. China is a representative country that uses carbendazim in agriculture. Since most of the herbal medicines imported to Korea are from China, decoctions made from herbal medicines may be contaminated with carbendazim [21,22].

According to the herbal medicine standard of Korea, carbendazim should be no more than 0.05 mg/kg in peony, 2.0 mg/kg in goji berry, 4.0 mg/kg in jujube, 10 mg/kg in Schisandra chinensis, 0.05 mg/kg in hemp, 2.0 mg/kg in raspberry, 0.5 mg/kg in ginseng, and 0.05 mg/kg in ginger. For agricultural products, the standard specifications are set for 123 items, e.g., eggplant, tangerine, potato, and mustard, and the permissible standard is set differently, from low to high concentration, depending on the item: less than 0.03 mg/kg of potato and less than 50 mg/kg of kale.

In this study, we detected 0.01 and 0.03 mg/kg in eight samples. According to a previous study, in the case of mushrooms, washing, drying, and heating can reduce carbendazim residues. In particular, there were no carbendazim residues after the boiling process [23]. Herbal medicines are also submitted to washing and drying processes, and herbal decoctions are the result of boiling herbal medicines; therefore, the carbendazim concentration should be detected at very low levels. We compared previous studies conducted in other countries because there are no pesticide residue standards for herbal decoctions in Korea.

Fan et al., analyzed 77 Fragaria and 74 Myrica rubra sold in Hangzhou, China. They detected prochloraz and carbendazim mostly with detection rates of 71.6% and 68.9% in Myrica rubra, and the mean concentration of carbendazim was 0.149 mg/kg (0.0110−1.02 mg/kg) [24]. In Zhang’s et al. [25] study, 99 Chuanxiong Rhizoma samples were analyzed. As a result, carbendazim and prometryn were the pesticides the most frequently detected, with a 100% detection rate. Carbendazim was found at 38.92 ± 83.68 μg/kg (0.38−343.55 μg/kg), which exceeded the Chinese Pharmacopoeia standard by 20 times [25]. In the study by Xiao, carbendazim was the most widely used pesticide (>85%) [26]. Since our study focused on herbal decoction and comparative studies targeted herbal medicine, which is the raw material for herbal decoction, it was not possible to individually compare the amount and the ratio of detection. However, it cannot be said that the level of pesticide residue contamination in herbal decoction is high.

This study has several limitations. The collection of 4 frequently used prescriptions in 10 external herbal dispensaries (EHD) was a valuable component of this study. However, the prescription names and composition of 40 samples could not be included in the thesis as the EHDs refused to disclose this information for business reasons. Consequently, it was not possible to provide them as a supplementary file. It is important to note that in Korea, national surveys are regularly conducted on the use of herbal medicines in TKM clinics, and these surveys suggest the frequently used prescriptions. Disclosure of the name and composition of the prescription may be possible in future studies if a standard prescription is selected and a dispensing request is made.

## 4. Materials and Methods

### 4.1. Sample Collection

From March to April 2020, the research team collected 40 herbal decoctions in a portable refrigerator from the 10 EHDs and used them for the experiment. Herbal decoction pouches were collected for 4 prescriptions used frequently in 10 EHDs. An EHD is a type of pharmacy that provides various types of herbal medicines to other TKM institutions in Korea [27].

### 4.2. Standards and Reagents

The pesticide standards used in the analysis were purchased from AccuStandard^®^ (New Haven, CT, USA) at a concentration of 1000 mg/L. Each pesticide standard was mixed and used, and acetonitrile (Merk, Rahway, NJ, USA) was used as a dilution solvent for each concentration. QuEChERS kits (mixed with anhydrous magnesium sulfate 4 g, sodium chloride 1 g, sodium citrate 1 g, disodium hydrogen citrate sesquihydrate 0.5 g, anhydrous magnesium sulfate 150 mg, and primary secondary amine 25 mg) were used with BEKOlut^®^ (Bruchmühlbach-Miesau, Germany) for pretreatment. The solvents used in the analysis were acetonitrile (hyper grade for LC-MS, Merk, Rahway, NJ, USA), formic acid (for LC-MS 98–100%, Merk, USA), and ammonium acetate (for mass spectrometry, Sigma-Aldrich, St. Louis, MI, USA).

A total of 113 pesticides subjected to GC-MS/MS analysis were prepared by mixing and diluting a standard product prepared at a concentration of 1000 mg/kg to a concentration of 5 mg/kg and then diluted to 5, 10, 20, 100, and 200 μg/kg using acetonitrile for analysis. A total of 207 pesticides subjected to LC-MS/MS analysis were prepared by mixing and diluting a standard product prepared at a concentration of 1000 mg/kg to a concentration of 2 mg/kg and then diluted to 1, 5, 10, 100, and 200 μg/kg using acetonitrile for analysis. Because the standard solution for analysis shows a matrix-induced chromate graphic response enhancement effect, in which the response value of the standard solution of pesticides increases as a result of the matrix, the extract of the nonpesticide decoction was mixed 1:1 with the standard solution for analysis.

### 4.3. Pretreatment of Samples

The pretreatment method for analyzing residual pesticides in decoction was as follows:(1)Precisely add 10 mL of decoction to a 50 mL centrifuge tube along with 10 mL of acetonitrile containing the internal standard (0.1 mg/kg triphenylphosphate), shake it, and extract it for 1 to 2 min.(2)Add 4 g of MgSO4, 1 g of NaCl, 1 g of trisodium citrate dihydrate, and 0.5 g of disodium hydrogen citrate sesquihydrate to the centrifuge tube of (1) and shake it for 1 min. After centrifugation (3000 rpm/min, 5 min) to separate the acetonitrile layer and the water layer, mix the acetonitrile layer with the buffer solution, filter it through a PTFE filter (0.2 μm), and analyze it by LC-MS/MS.(3)In the case of GC-MS/MS, add 1 mL of acetonitrile extract from (2) to a powdered solid phase extraction tube containing 150 mg of MgSO_4_ and 25 mg of PSA, shake it for 1 min, and centrifuge it (10,000 rpm/min, 2 min). Then, analyze it by GC-MS/MS.

In this study, the possibility of a multicomponent simultaneous analysis method using LC-MS/MS and GC-MS/MS for pesticide residue analysis in a decoction was confirmed by applying a modified pretreatment method of the QuEChERS sample pretreatment method (Figure 3).

#### Analysis of Instrument and Instrument Conditions

A total of 320 pesticide residue analysis methods of decoction were extracted and purified using salt-containing acetonitrile/powder phase solid phase extraction (QuEChERS); 207 species were analyzed by LC-MS/MS, and 113 species were analyzed by GC-MS/MS.

The MRM conditions for the 207 pesticides subjected to LC-MS/MS analysis and the 113 pesticides subjected to GC-MS/MS analysis are provided in Appendix A. The slope and correlation (R^2^) of the analytical pesticide standard solution calibration curve of LC-MS/MS and GC-MS/MS are presented in Appendix A.

### 4.4. Validation of the Test Method

The validity of the analysis method was verified using specificity, linearity, accuracy, precision, detection limit, and quantification limit.

Accuracy was confirmed by a recovery experiment for the standard solution, and the standard treatment concentrations were 10, 50, and 100 μg/kg, including the quantification limit for each pesticide component. the results were confirmed after three repetitions.

The detection and quantification limits were calculated using an analytical instrument program after 7 repeated measurements of 10 μg/kg concentration based on a calibration curve prepared at concentration levels of 1 to 200 μg/kg.

## 5. Conclusions

A total of 320 pesticide residues in 40 decoctions were analyzed using an improved LC-MS/MS and GC-MS/MS analysis. As a result of the monitoring, carbendazim was detected at 0.01 and 0.03 μg/g in eight samples, and no pesticide was detected in the other herbal decoctions. In addition, this study verified the specificity, linearity, accuracy, detection limit, and quantification limit of pesticides using improved LC-MS/MS and GC-MS/MS analysis methods compared with existing methods (multiclass pesticide multiresidue methods). Therefore, our results provide a framework for pesticide residue management in countries with traditional medicines.

## Figures and Tables

**Figure 1 molecules-28-03343-f001:**
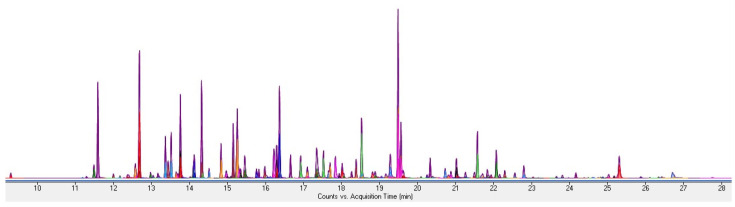
GC-MS/MS standard solution chromatogram. GC: gas chromatography; MS: mass spectrometry.

**Figure 2 molecules-28-03343-f002:**
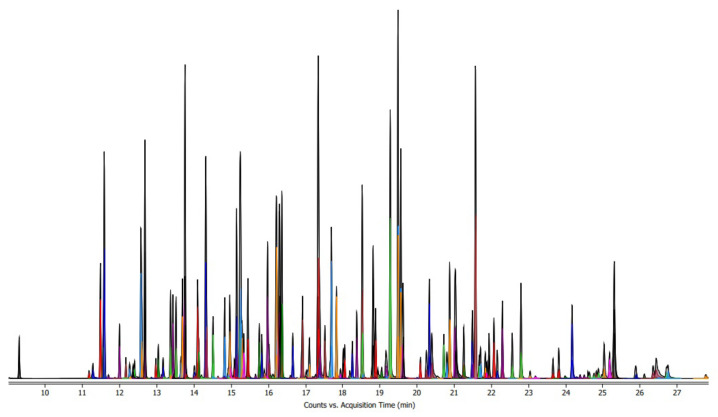
LC-MS/MS standard solution chromatogram. LC: liquid chromatography; MS: mass spectrometry.

**Figure 3 molecules-28-03343-f003:**
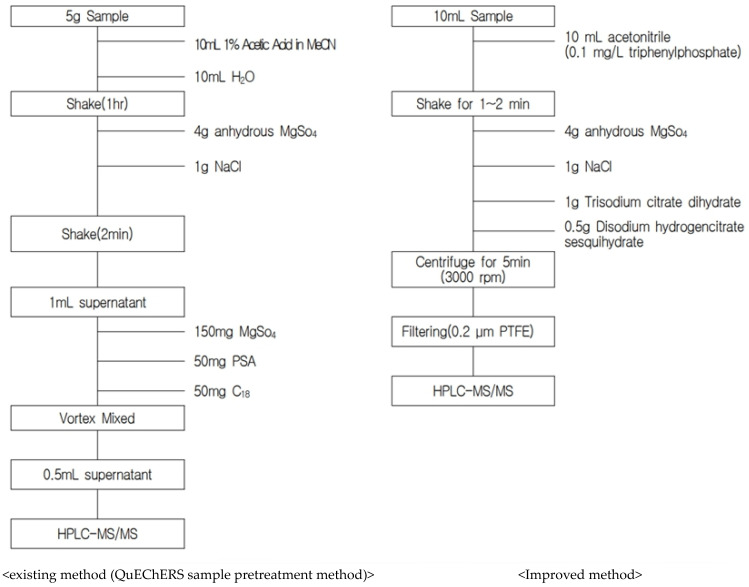
Pretreatment method for analyzing residual pesticides of herbal decoctions. PTFE: polytetrafluoroethylene; GC: gas chromatography; LC: liquid chromatography; MS: mass spectrometry.

## Data Availability

The data will be made available upon reasonable request.

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
