# Peer review of "Chromatographic Method for Monitoring of Pesticide Residues and Risk Assessment for Herbal Decoctions Used in Traditional Korean Medicine Clinics"

_molecules, 2023, doi:10.3390/molecules28083343_

Round 1

Reviewer 1 Report

The manuscript by Se-Mi Kang and colleagues monitored of pesticide residues in herbal decoctions used in traditional Korean medicine. This study tested 40 samples of herbal decoctions and analyzed for the presence of 320 pesticide residues using liquid chromatography and gas chromatography coupled with tandem mass spectrometry. In my opinion, there are viewpoints in the manuscript that are not clear and too generally. The author and colleagues should consider rewriting abstract, introduction and some tables and figures. Insert the list of tested herbal decoctions, and explain, Why did you choose these 40 samples? On based…. In introduction authors write only methods and risk assessment of pesticide, no mention about choosing tested samples. Are these herbals most used in the Traditional Korean Medicine? In the manuscript are many tables, unnecessarily. Conditions of analysis by GC/LC-MS/MS will be better write in the text, no in the table. What is the Galgeun-tang? Did authors use of internal standard or standard addition? Figure 3: The dependence of the calibration line should start from zero. The x-axis is from -10. Why? Title of y-axis....Responses.....of what? I recommend remodeling this Figure. Why are there abbreviations of GC, LC and MS everywhere? Abbreviations should be given in the list of abbreviations or at first mention. I do have several major comments on this submitted manuscript (above), which should be carefully addressed.

Author Response

We appreciate the time given and efforts made by the editor and referees in reviewing this paper. Please see the attachment.

Reviewer 2 Report

The manuscript entitled “Monitoring of Pesticide Residues and Risk Assessment for Herbal Decoctions used in Traditional Korean Medicine Clinics” aims to evaluate residual pesticide levels in herbal medicines used in traditional Korean medicine clinics. A total of 40 samples of herbal decoctions were collected and analyzed for the presence of 320 pesticide residues using liquid chromatography-tandem mass spectrometry and gas chromatography-mass spectrometry. The manuscript is well written in general. The topic is interesting. It fits well with the scope of the Journal.

The abstract is good.

Introduction: lines 60-72 are unnecessary. The results in Table 1 should be transferred to text only since most of the pesticides are not detected.

The discussion is good.

Materials and Methods are detailed enough.

The conclusion needs to be expanded.

Author Response

(The authors gave the same response as above.)

Reviewer 3 Report

The authors presented a manuscript describing an approach using a LC and GC techniques coupled to MS detectors for the analysis of over 300 pesticides.

Before publishing please consider following comments:

1) please add the techniques used in the title of the paper

2) you state that the aim of the study is to use a single test method to supplement the pretreatment process. Perhaps comparing your results obtained with the suggested procedure to already established methods would be valuable and demonstrate that the method can be used instead.

3) Please omit Figures 3 and 4 and put them in the Supplement. They are to basic. Just state the obtained equation and the R2.

4) Put the results of the method validation, i.e. accuracy in a form of a Table so it is easier to follow.

5) How do the numbers of target concentrations in Table 2 and Table 3 correspond to the graphs in the Figures 3 and 4?

6) Compare the obtain LOD/LOQ leveles (lines 148-150) to expected levels of the pesticides of interest in your samples, to toxic levels, and to legally permitted levels?

Please state the concentrations in the ug/g levels as well. Otherwise it is impossible to follow the results presented in the Table 1.

7) How did you test the specificity of the method? Probably it was selectivity?

8) Table 4 uses the whole page of the manuscript. You can just write this in sentences and it will use 2 paragraphs.

9) are the suggested LC and GC methods novel, or are you using some previously described methods?

10) Why did you use the mentioned approach to calcute LOD/LOQ? Please describe what the software did?

11) How does the range 1-200 ug/L (line 300) correspond to table 3?

Author Response

(The authors gave the same response as above.)

Round 2

Reviewer 3 Report

The authors have made significant changes to the document according to reviewers' comments.

I don't understand the answer to comment 5, since you hae replaced the accuracy of the method with the description of the preatment procedure in a form of a scheme. Where are the accuracy results?

Comment 11 does not answet the question about the novelty of LC and GC procedures. You are refereing to the pretreatment method. Please just refer to the chromatographic part of the methods you used, since chromatography is in the title of the manuscript.

Author Response

(The authors gave the same response as above.)
